# Genetic Analysis and Operative Outcomes in Patients with Oncogene-Driven Advanced NSCLC Treated with Cytoreductive Surgery as a Component of Local Consolidative Therapy

**DOI:** 10.3390/cancers13112549

**Published:** 2021-05-22

**Authors:** Byung Jo Park, Hyo Sup Shim, Chang Young Lee, Jin Gu Lee, Hye Ryun Kim, Sang Hoon Lee, Min Hee Hong, Seong Yong Park

**Affiliations:** 1Department of Thoracic and Cardiovascular Surgery, Yonsei University College of Medicine, Seoul 03722, Korea; BZPARK@yuhs.ac (B.J.P.); CYLEECS@yuhs.ac (C.Y.L.); CSJGLEE@yuhs.ac (J.G.L.); 2Department of Pathology, Severance Hospital, Yonsei University College of Medicine, Seoul 03722, Korea; SHIMHS@yuhs.ac; 3Division of Medical Oncology, Department of Internal Medicine, Yonsei Cancer Center, Severance Hospital, Yonsei University College of Medicine, Seoul 03722, Korea; nobelg@yuhs.ac; 4Division of Pulmonology, Department of Internal Medicine, Yonsei University College of Medicine, Seoul 03722, Korea; cloud9@yuhs.ac

**Keywords:** advanced lung cancer, cytoreductive surgery, local consolidation, oncogenic-driven, oligometastasis

## Abstract

**Simple Summary:**

The efficacy of the local consolidative therapy in lung cancer has been reported previously, however, those studies described the utility of the radiation therapy rather than of surgery. Furthermore, those studies included either no or few patients with oncogene-driven NSCLC, which has distinct biological properties and treatment options. Surgery is the most reliable method of tumor removal that allows detailed examinations of resected tissue, such as comprehensive genetic analysis. This retrospective study for the first time reports the operative outcomes and the benefit of cytoreductive surgery in TKI-treated cases of oncogene-driven locally advanced or metastatic NSCLC in parallel with the genetic analysis of the tumor.

**Abstract:**

Most patients with oncogene-driven advanced non-small cell lung cancer (NSCLC) demonstrate recurrence because of the developing targeted therapy resistance. In this retrospective study, we assessed the efficacy of surgical local consolidative treatment by analyzing the operative outcomes and genetic data in 44 patients who underwent pulmonary resection for stage IIIB/C–IV NSCLC after targeted therapy. The initial mutations were in the *EGFR* (*n* = 32), *ALK* (*n* = 11), and *ROS1* (*n* = 1) genes. The median interval from the initiation of tyrosine kinase inhibitor (TKI) therapy immediately before the surgery to the actual operation was 9.8 months. Operative mortality was absent. Four patients showed complete remission. The median follow-up period after TKI therapy initiation was 23.1 months. The Kaplan–Meier survival analysis showed that the 2-year failure-free survival and overall survival rates from the initiation of TKI were 70.8% and 95.0%, respectively. During the follow-up period, two patients died and 15 suffered from disease progression. Among the 32 patients with *EGFR* mutations, 12 showed additional mutations, and targeted agents were replaced in nine patients after the operation. We conclude that pulmonary resection for advanced NSCLC after targeted therapy is feasible, and the surgical specimens could be used for planning further targeted therapy.

## 1. Introduction

Agents targeting proteins encoded by mutated *EGFR*, *ALK*, and *ROS1* genes significantly improve the prognosis of advanced oncogene-driven non-small cell lung cancer (NSCLC) [1,2]. Nonetheless, most patients eventually experience disease progression because of the developing resistance to such targeted agents [3]. Approximately 60% of patients develop first disease progression at the primary cancer sites [4,5]. Therefore, following the initial response to targeted treatment, local consolidative therapy aimed at the residual disease could potentially overcome the resistance and improve survival.

Randomized trials have provided evidence that this strategy could be effective in advanced lung cancer [6,7,8]. In oligometastatic NSCLC, randomized studies have shown that local consolidative therapy leads to higher progression-free (PFS) and overall survival (OS) benefits compared with those afforded by either maintenance therapy or observation. However, these studies included either no or few patients with oncogene-driven NSCLC, which has distinct biological properties and treatment options. A recent phase II study of residual metabolic disease showed that local ablative therapy after treatment with targeted agents leads to better PFS [9]. However, that study had few patients and, as in previous studies, radiation therapy was chosen as local consolidative therapy rather than surgery.

Intratumoral heterogeneity, tumor cell evolution after targeted therapy, and pharmacokinetic failure are the predominant causes of unsuccessful treatment outcome [10,11]. Surgery is the most reliable method of tumor removal that allows detailed examination of resected tissue, such as comprehensive genetic analysis and cytological tests. However, many important questions must be addressed before proceeding to further prospective studies. In particular, the safety of surgery in the local consolidation setting has to be prioritized.

Here, we assessed the operative outcomes of cytoreductive surgery with local consolidative purpose and genetic information of surgical specimens in patients with oncogene-driven advanced NSCLC after targeted therapy.

## 2. Patients and Methods

### 2.1. Patients

This study was a retrospective review of a prospective lung cancer database at our institution, using data collected between March 2018 and July 2020. The study was reviewed and approved by the Institutional Review Board of the Yonsei University College of Medicine (IRB No. 4-2020-1227). The inclusion criteria required patients to have the following: (1) histologically proven oncogene-driven NSCLC; (2) stage IIIB, IIIC, or IV advanced NSCLC; and (3) to have undergone surgical resection as a form of local consolidative therapy after treatment with tyrosine kinase inhibitors (TKIs).

Indication of operation was decided at a meeting of a multidisciplinary team that consisted of thoracic surgeons, medical oncologists, pulmonologists, radiologists, and radiation oncologists. We selected patients with good performance status (ECOG 0 to 1), medically fit patients without problems that might increase the risk of surgery, and patients with no or stable remnant metastatic lesions or those treated with other local consolidative therapy for the remaining lesions after surgery. Patients were classified into two subgroups based on the intent of surgery, one with residual lesions and the other with progressive lesions after TKI therapy started immediately before the surgery.

We performed anatomic resection with complete mediastinal lymph node dissection in most patients. Some patients underwent limited resection, depending on the situation during surgery. In patients with cervical or abdominal lymph node metastasis, the lymph node dissection at those areas was conducted concomitantly. TKI therapy was continued in perioperative periods. The complications were graded based on the Clavien–Dindo classification [12]. After the operation, the patients were followed up using chest and abdomen computed tomography every 4 months for detecting the recurrence.

### 2.2. Molecular Analyses

To detect *EGFR* mutations, peptide nucleic acid-mediated real-time PCR was performed using the PANA Mutyper EGFR Kit (PANAGENE, Daejeon, Korea). To identify *ALK* rearrangements, the VENTANA *ALK* (D5F3) CDx Assay (Ventana Medical Systems, Tucson, AZ, USA) was performed. *ROS1* rearrangements were detected with real-time PCR using the *ROS1* Gene Fusions Detection Kit (AmoyDx, Xiamen, China). Their detection was also possible by next-generation sequencing, which will be described as follows.

### 2.3. Next-Generation Sequencing (NGS)

Targeted DNA sequencing was performed using TruSight™ Oncology 500 (Illumina, San Diego, CA, USA). The TruSight™ Oncology 500 DNA panel was designed to detect 523 cancer-related genes with potential single nucleotide variants and indels, as well as 59 genes with potential amplifications. Briefly, 40 ng of formalin-fixed paraffin-embedded (FFPE) tissue-derived DNA was extracted using the QIAGEN AllPrep FFPE Kit (Qiagen, Hilden, Germany). After hybridization capture-based target enrichment, paired-end sequencing (2 × 150 bp) was performed using a NextSeq sequencer (Illumina). Variants with a total depth of at least 100× and variant allele frequency of at least 1% were included for analysis. Variant interpretation was based on the recommendations of the Association for Molecular Pathology, American Society of Clinical Oncology, and College of American Pathologists [13]. Actionable genetic alterations were stratified into one of the four levels based on the OncoKB website (http://www.OncoKB.org, accessed 1 November 2020).

### 2.4. Statistical Analysis

Descriptive statistics were used to illustrate patient characteristics and outcomes. Categorical variables are presented as frequencies and percentages; continuous variables are expressed as the median with the range or interquartile range (IQR). Student’s *t*-test and Mann–Whitney test, depending on the normality of distribution, and the *χ*^2^ test or Fisher’s exact test were used to compare continuous and categorical variables, respectively. Based on the criteria proposed by Travis et al. [14], MPR (major pathologic response) is defined historically as 10% or less residual viable tumor following preoperative treatment, and CR (complete pathologic response) is defined as no viable tumor following preoperative treatment. Treatment failure-free survival (FFS) duration was defined as the time from the initiation of TKI therapy immediately before the surgery to the earliest occurrence of disease progression, end of treatment because of adverse events, or death. The duration of PFS was defined as the time from surgery to disease progression or death from any cause. The duration of OS was defined as the time from the initiation of TKI therapy immediately before the surgery to death from any cause. Patients lost to follow-up were censored at the time of the last contact. Actuarial survival curves were estimated using the Kaplan–Meier method. Analyses were performed using SPSS version 25.0 (IBM Corporation, Somers, NY, USA) and R software (version 4.0.3) with the “survival” and “ComplexHeatmap” packages (R Foundation for Statistical Computing, Vienna, Austria).

## 3. Results

### 3.1. Baseline Characteristics and Treatments Prior to Surgery

From March 2018 to July 2020, 44 patients received pulmonary resection and mediastinal lymph node dissection for stage III B–C or IV A–B NSCLC after TKI therapy (Table 1). The median age of patients was 59 years (range: 28–75 years), and 15 patients (34.1%) were males. The initial stages were IIIB (*n* = 4), IIIC (*n* = 1), IVA (*n* = 15), and IVB (*n* = 24). The initial metastasis sites were the brain (*n* = 10), bone (*n* = 6), lymph nodes (*n* = 6), intrathoracic organs (*n* = 5), and multiple organs (*n* = 17). At the time of diagnosis, 22 patients (50%) had combined brain metastasis. The initial mutations were in the *EGFR* (*n* = 32), *ALK* (*n* = 11), and *ROS1* (*n* = 1) genes.

The preoperative disease status and treatment of these patients are presented in Table 2.

The median interval from the initiation of TKI therapy immediately before surgery to actual operation was 9.8 months (range: 2.2–25.4 months). In regard to best responses, 41 patients (93.2%) had partial response, and three patients (6.8%) had stable disease after the commencement of TKI therapy immediately before surgery according to RECIST criteria [15]. Among them, eight and two patients showed disease progression in the primary site and metastatic sites, respectively. Among the 32 patients with *EGFR* mutations, 15 patients underwent surgery during gefitinib treatment. Of the 12 patients treated with osimertinib before surgery, six received it as their palliative first-line therapy and the other six had it as their second-line therapy based on the acquired T790M mutation. In patients treated with the later line osimertinib, the first-line treatments were afatinib (*n* = 3), gefitinib (*n* = 1), erlotinib (*n* = 1), and AZD3759 (*n* = 1). In the 11 *ALK*-positive patients, the most common regimen before surgery was alectinib (*n* = 7), including five first-line alectinib cases. The treatment courses of individual patients are shown in Figure 1.

In 21 patients (47.7%), other metastatic lesions were treated with surgery (*n* = 4, 9.1%), radiation therapy (*n* = 15, 34.1%), or a combination thereof (*n* = 2, 4.5%) prior to pulmonary resection. At the primary lesion, 6 patients (13.6%) had partial response, 30 (68.2%) presented with stable disease, and 8 (18.2%) experienced disease progression. At the metastatic site, 19 patients (43.2%) had no radiologic evidence of disease, 2 (4.5%) had partial response, 21 (47.7%) had stable disease, and 2 (4.5%) experienced disease progression. Of the two patients with disease progression in the metastatic lesion, one underwent additional wedge resection of the metastatic lung lesion during surgery, whereas the other patient underwent additional radiation therapy of the metastatic bone lesion after surgery. Thirty-four patients (77.3%) received surgery for their residual lesion, whereas 10 patients (22.7%) had it for their progressive lesion.

### 3.2. Surgical Outcomes

The operative outcomes are shown in Table 3.

Lobectomy was performed in 37 patients (84.1%), sub-lobar resection in 5 patients (11.4%), and bi-lobectomy in 2 patients (4.5%). The surgical approaches were video-assisted thoracoscopic surgery (VATS) (*n* = 37, 84.1%), thoracotomy (*n* = 2, 4.5%), and conversion to thoracotomy from VATS (*n* = 5; 11.4%). The reason for the conversion was the difficulty of dissecting anthracosis or fibrotic lymph nodes between the bronchus and pulmonary artery. Two of these were accompanied by pulmonary artery injury during dissection. The following combined surgical procedures were performed in 20 patients: cervical lymph node dissection (*n* = 7, 15.9%), abdominal lymph node dissection (*n* = 1, 2.3%), additional pulmonary wedge resection (*n* = 4, 9.1%), en-bloc resection of the adjacent lobe (*n* = 5, 11.4%), pericardial biopsy (*n* = 1, 2.3%), pleural biopsy (*n* = 1, 2.3%), and total thyroidectomy with central neck node dissection for concomitant thyroid cancer (*n* = 1, 2.3%). The median operation time was 108.5 min (range: 67–308 min). Most of the patients (*n* = 41, 93.2%) had a relatively small blood loss of less than 200 mL, and only three patients (6.8%) lost more than 200 mL of blood. Complete resection was achieved in 41 patients (93.2%). Only one patient (2.3%) with intraoperative bleeding stayed in the intensive care unit for 1 day, and the median postoperative hospital stay was 5 days (range: 3–32 days). The following complications were observed in 15 patients (34.1%): prolonged air leakages (*n* = 5, 11.4%), chyle leakages (*n* = 3, 6.8%), vocal cord palsies (*n* = 2, 4.5%), bronchopleural fistula (*n* = 1, 2.3%), acute kidney injury (*n* = 1, 2.3%), acute lung injury (*n* = 1, 2.3%), pneumonia (*n* = 1, 2.3%), and pneumothorax (*n* = 1, 2.3%). There were four complications (9.1%) of Clavien–Dindo classification grade 3 that required additional procedures. Operative mortality was absent.

### 3.3. Pathologic Analysis of Surgical Specimens

Pathologic analyses and the postoperative disease status are described in Table 4. Twenty-three patients (52.3%) achieved radiologic no-evidence-of-disease status after surgery. Furthermore, 10 patients (22.7%) achieved MPR, including 5 patients CR (11.4%), at the primary site regardless of the mutation or treatment regimen. Among the 32 patients with *EGFR* mutations, seven (21.9%) achieved MPR. Among the 11 patients with *ALK* fusions, MPR was observed in seven patients (63.7%), including four patients (36.4%) with complete pathologic response. The one patient (100%) with *ROS**1* fusion showed complete pathologic response at the primary lesion.

### 3.4. NGS and Mutation Profiles

We conducted NGS and PCR mainly in the surgical tumor samples from *EGFR*-mutant patients (Figure 2); their mutation profile changes are shown in Figure 3. 

Among the 18 patients with sensitizing *EGFR* mutations treated with first- or second-generation EGFR-TKIs, 11 patients (61.1%) developed the T790M mutation, and four samples exhibited a discrepancy between PCR and NGS results (in three cases, T790M mutation was revealed by PCR only and in one case, it was reported by NGS only). Nine out of 11 patients with T790M mutations were switched to osimertinib either immediately after surgery (*n* = 3) or when radiologic disease progression became evident (*n* = 6). Out of the six patients treated with second-line osimertinib before surgery based on the acquired T790M mutation in preoperative additional biopsy mutation analysis results, three lost their T790M mutation and one developed the *EGFR* C797G mutation. In addition, five had *TP53* and *RB1* mutations at the time of surgery. Among them, one patient (P3) experienced SCLC transformation 1 year after the surgery; he had *TP53* c.2369C>T (or p.V173M) mutation and *RB1* c.1332+1G>C mutation (splice variant at protein level). Another patient (P16) experienced the same event 2 years after the surgery; he retained a *TP53* indel mutation (p.MG243IC) and an *RB1* frameshift mutation (p.V654fs*4).

### 3.5. Survival Analysis

The median follow-up period after the initiation of TKI therapy immediately before the surgery and the postoperative follow-up period were 23.1 months (range: 4.8–41.8 months) and 12.0 months (range: 2.6–32.9 months) months, respectively. The Kaplan–Meier survival analysis showed that the 2-year FFS and OS rates from the initiation of TKI immediately before the surgery were 70.8% and 95.0%, respectively (Figure 4A,B). In patients with *EGFR* mutations, these parameters were 58.5% and 93.0%, respectively, and they were 100.0% in patients with *ALK* and *ROS1* fusions. The 1-year PFS and OS rates from the operation were 64.8% and 94.8%, respectively (Figure 4C,D). The Kaplan–Meier survival curves according to the intent of surgery and pathological response at the primary lesion are shown in Figure 5.

During the follow-up periods, 15 patients experienced disease progression in the lung (*n* = 6), brain (*n* = 4), bone (*n* = 2), pleura (*n* = 1), and other sites (*n* = 2). Two patients died during the follow-up period, one patient died 7.9 months postoperatively because of cancer progression, and one other patient died of acute coronary syndrome at 2.6 months postoperatively.

## 4. Discussion

The initial systemic therapy may lead to stable or responsive disease, but the remaining tumors may contain treatment-resistant malignant cells that are not eliminated by the maintenance therapy. These remaining drug-tolerant persistent tumors may cause the subsequent metastatic spread even if they do not have radiographic progression [16]. Therefore, local consolidative treatment may delay clinical disease progression and improve prognosis by removing such drug-tolerant persister cells. However, radiation therapy has predominantly been used as a local consolidative treatment in the published trials [6,7,8,9]. A recent study in patients with advanced EGFR-mutant NSCLC reported that TKIs plus thoracic stereotactic body radiation therapy (SBRT) significantly extends PFS with tolerable toxicity [17]. They reported median PFS was 19.4 months in the TKIs plus SBRT group. Our group has conducted surgical resection as a local consolidative treatment in advanced NSCLC with driver mutations. In the present study of a limited set of patients with relatively short follow-up periods, the 2-year FFS and OS rates were 70.8% and 95.0%, respectively, whereas median OS and FFS had not yet been reached. Although it is difficult to directly compare our results to the previous reports, our findings show a promising prognosis with tolerable surgical morbidity and tumor samples for genetic analysis. The rationale for pulmonary resection and its benefits over consolidative radiotherapy is as follows: (1) TKI-treated patients ultimately develop resistance to the drugs, a condition for which there is no established treatment; (2) primary lung lesion is the most common resistant site; (3) surgery is the most reliable method of tumor removal that enables accurate staging and treatment through mediastinal lymph node dissection; and (4) surgical tumor samples provide information about tumor heterogeneity and mutational evolution to guide subsequent treatment. Surgical outcomes after TKI use in advanced NSCLC with driver mutation have been reported for a small group of patients [18]. This retrospective study for the first time reports the operative outcomes and the benefit of cytoreductive surgery in TKI-treated cases of oncogene-driven locally advanced or metastatic NSCLC in parallel with the genetic analysis of the tumor.

Our analysis of the surgical outcomes after TKI therapy demonstrated that surgical intervention was feasible and safe; operative mortality was absent and operative morbidities were observed only in 15 cases (34.1%). The four complications (9.1%) above grade 3 were chylothorax, vocal cord palsy, bronchopleural fistula, and delayed pneumothorax. Minimally invasive resection was carried out in 37 patients (84.1%). The conversion rate to the open thoracotomy occurred in five patients (11.4%). In most patients, it was difficult to dissect the lymph node and the bronchus or pulmonary artery, and two patients had bleeding because of pulmonary artery injury. Owing to the response of cancer cells to the TKI, previously metastatic lymph nodes became fibrotic, which complicated the operation technically, especially around the pulmonary artery (Figure 6). Our conversion rate was higher than the thoracotomy conversion rate in simple VATS lobectomy in NSCLC [19], but it is acceptable when compared to the results in another neoadjuvant series [20]. Operational time, blood loss, and hospital stay were longer than those in simple pulmonary resection, but generally acceptable. We did not stop TKI therapy during the perioperative periods, in contrast to what is usually done for cytotoxic chemotherapy agents. There is no reason to believe that TKI therapy itself caused any of the observed complications. Another advantage of surgical resection was the possibility to examine the thoracic cavity. For example, pleural seeding, which was not noticed at the preoperative imaging, was thus observed in two patients.

An important advantage of surgery over radiation therapy is the possibility to obtain tissue samples for pathologic examination and genetic analysis. This enables the evaluation of TKI efficacy by determining the fraction of the remnant viable tumor, MPR, and pathologic complete response, which, in turn, makes the prognosis more accurate and helps in establishing further treatment plans. Among the 44 patients, five (11.4%) showed a complete response and 10 (22.7%) showed MPR at the primary lesion. As the initial tumor stages were mostly stage IV or late stage III, TKI therapy was continued even in patients who showed complete response. Even though there have been no recurrences in the patients who had a complete response in the set follow-up periods, whether complete response or MPR correlated with improved survival has to be established by the longer follow-up. It has been reported that EGFR-TKIs are cytostatic rather than cytotoxic agents, which do not eradicate micrometastatic tumor cells even after a marked clinical response [21]. Interestingly, all our patients who showed the complete response at the primary and metastatic lesions had *ALK* rearrangement. This suggests that NSCLC with *ALK* fusion is relatively homogenous and may display features of oncogene addiction, which, in turn, may explain higher median survival of patients with *ALK* fusions treated with ALK-TKIs compared to that of patients with *EGFR* mutations treated with EGFR-TKIs [8,22]. These findings warrant further studies of the biomarkers of complete response to *ALK* inhibitors.

Mechanisms of acquired resistance to front-line and later-line osimertinib are similar, except the absence of the T790M mutation after the former regimen [23]. In line with previous reports [24], we observed that half of the patients with the *EGFR* T790M mutation lost it after later-line osimertinib treatment. We also observed inconsistent PCR and NGS results, which may reflect the difference in the performance of the testing method and/or tumor heterogeneity. Considering that NGS sensitivity is higher than that of PCR in general, and that one and three patients were found to have T790M mutation only using NGS and only using PCR, respectively, this discrepancy was likely because of tumor heterogeneity. Our data showed the trend that patients with *EGFR* mutations that harbored fewer concurrent mutations had better response to EGFR-TKIs (Figure 2A), which was consistent with the recent reports [25,26,27]. The objective response rate in patients without concomitant mutations was significantly higher (77% vs. 44%) [25]. In addition, the mutational analysis of postoperative tumor tissue may reveal acquired resistance mechanisms and guide subsequent treatment, as was demonstrated in our data set. Nine patients had started osimertinib because of the revealed *EGFR* T790M mutation, and two patients were enrolled in an MET inhibitor trial based on the presence of *MET* amplification. The occurrence of *TP53* and *RB1* mutations during the EGFR-TKI treatment predicted SCLC transformation in two patients. The inactivating *TP53* and *RB1* mutations were detected 1 or 2 years before clinical progression, which supported the notion that EGFR-TKI-resistant SCLC cells branches out at the early stage from the adenocarcinoma clones [28].

Regarding the diagnostic accuracy, the methods that we used do not capture every oncogenic alteration in EGFR mutation and ROS1 fusion. However, detecting EGFR mutations, the PANAMutyper assay can capture sensitizing mutations (19del and L858R), resistant mutation (T790M) and major uncommon EGFR mutations (S768I, G719X, and L861Q), and is highly concordant to the Roche cobas^®^ EGFR version 2, which is approved by the U.S. Food and Drug Administration [29]. In addition, ROS1 fusion genes detectable by the AmoyDx ROS1 Gene Fusions Detection Kit account for approximately 83% of all ROS1 gene fusions [30]. However, considering that all of the patients in the study had oncogenic genetic alteration at baseline, the analytical sensitivity of the PCR-based test can be said to be 100% based on the oncogenic (driver) mutation in the preoperative sample.

This study had some limitations. First, it was a single-center retrospective study with few patients. Second, NGS data from the tissue before TKI treatment were not available; therefore, whether the revealed genetic alterations had been already present at baseline remained unknown. Third, we did not perform a liquid biopsy before and after the TKIs. If performed, it could help to capture tumor heterogeneity. Lastly, the effects of surgery on OS must be evaluated with longer follow-up periods and further prospective clinical trials. Despite these limitations, this study is valuable, as it is the first report on the outcomes and pathologic results of cytoreductive surgery in patients with oncogene-driven locally advanced or metastatic NSCLC. Many questions, such as operation timing (early vs. late operation), selection of the proper indication, including the suitable biomarker, extent of surgical resection (anatomic resection vs. sub-lobar resection), and continuation of TKI, remain to be resolved in further studies.

## 5. Conclusions

In conclusion, pulmonary resection for advanced NSCLC after targeted therapy is feasible, and the surgical specimens obtained could be used for further planning of targeted therapy. The long-term benefits of pulmonary resection on survival after targeted therapy must be studied in future trials.

## Figures and Tables

**Figure 1 cancers-13-02549-f001:**
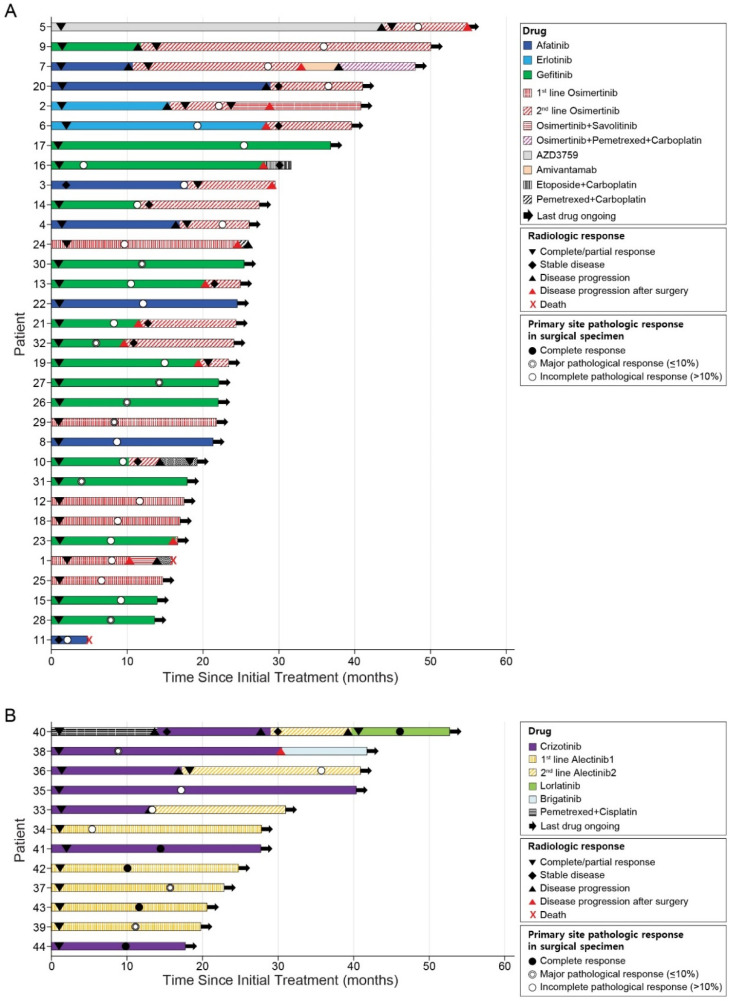
Survival of patients with oncogene-driven advanced NSCLC after the initial treatment. (**A**) The swimmer plot of the 32 patients with *EGFR* mutations. (**B**) The swimmer plot of the 12 patients with *ALK* or *ROS1* fusions.

**Figure 2 cancers-13-02549-f002:**
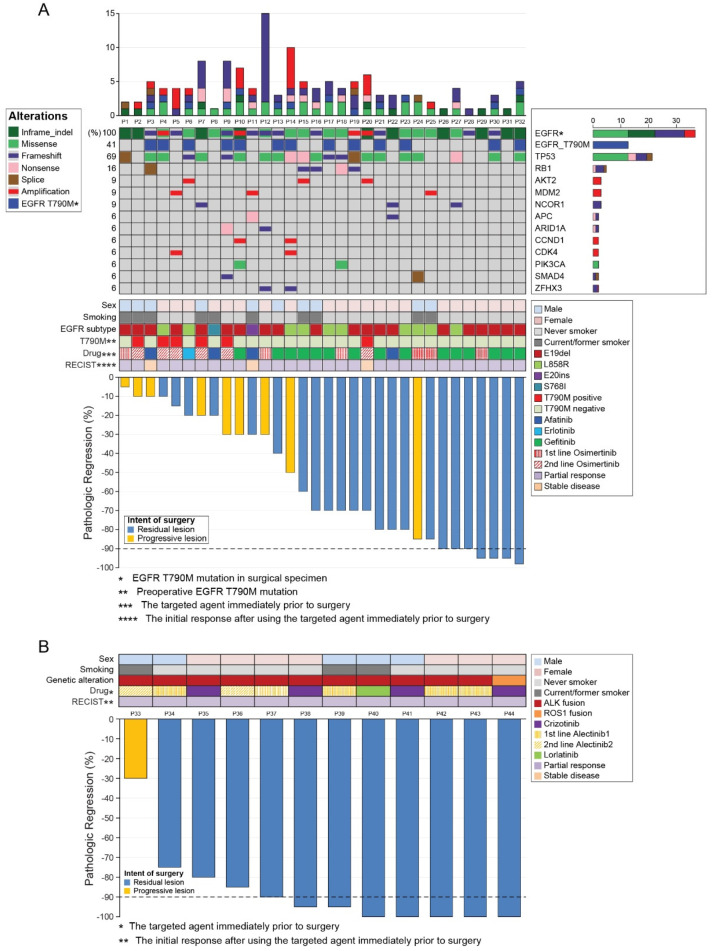
Landscape of mutation profiles and pathological responses at the primary lesion. (**A**) Landscape of mutation profiles analyzed using next-generation sequencing of samples from the 32 patients with *EGFR* mutations. The X-axis represents individual samples, and the Y-axis represents individual mutated genes. The bar graph on the right shows the mutation frequency of each mutated gene in the 32 samples. The top column represents tumor mutation burden. Different colors and notes at the left represent mutation types. Middle rows indicate sex, smoking status, *EGFR* mutation subtype, presence of the T790M mutation before surgery, targeted agent immediately before surgery, and best response after using the targeted agent immediately before surgery. The Waterfall chart at the bottom shows the pathologic response at the primary lesion of the 32 patients with *EGFR* mutations. (**B**) The Waterfall chart shows the pathologic response at the primary lesion of the 12 patients with *ALK* or *ROS1* fusions. Upper rows indicate sex, smoking status, genetic alteration, targeted agent immediately before surgery, and best response after using the targeted agent immediately before surgery. RECIST: Response Evaluation Criteria in Solid Tumors.

**Figure 3 cancers-13-02549-f003:**
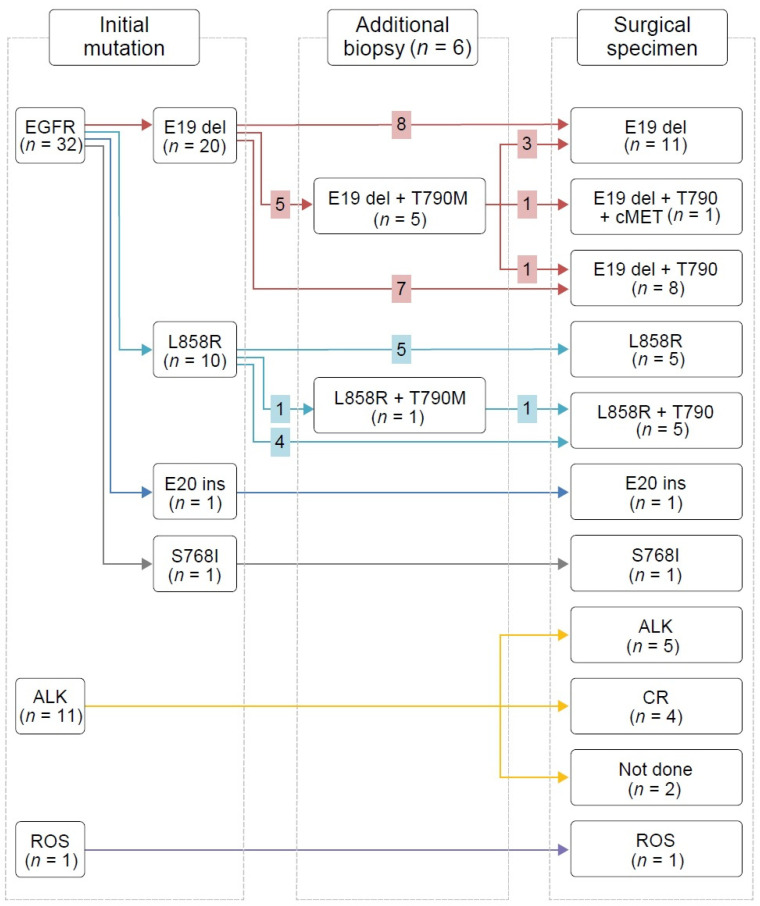
Mutation profile changes.

**Figure 4 cancers-13-02549-f004:**
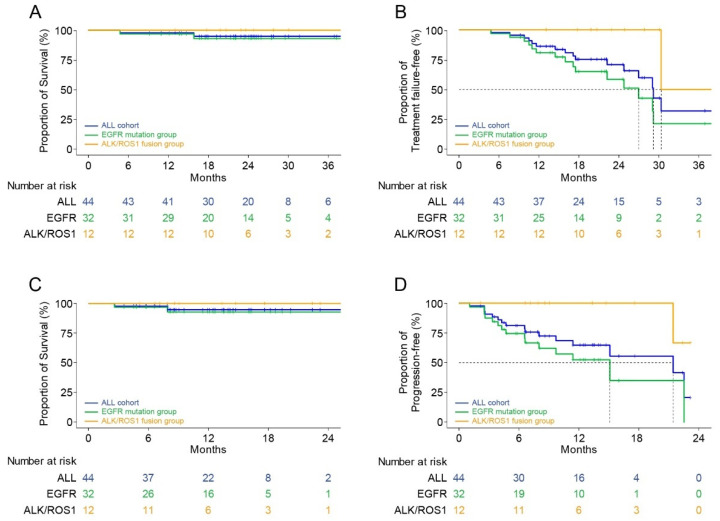
Kaplan–Meier survival curves according to the mutation. (**A**) Overall survival after using the targeted agent immediately before surgery. (**B**) Treatment failure-free survival after using the targeted agent immediately before surgery. (**C**) Overall survival after surgery. (**D**) Progression-free survival after surgery.

**Figure 5 cancers-13-02549-f005:**
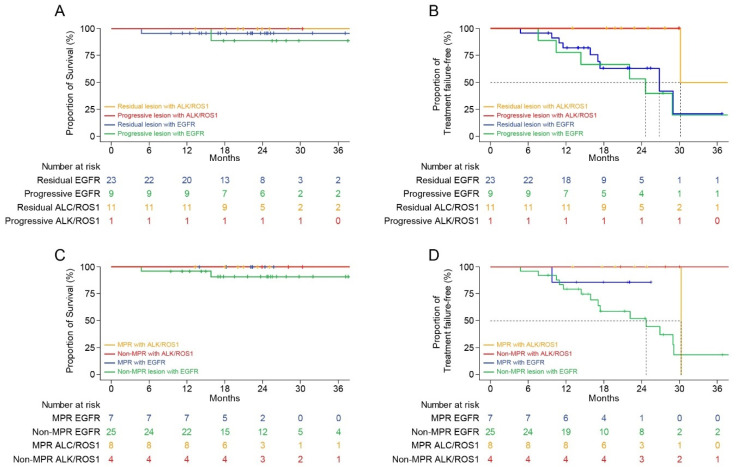
Kaplan–Meier survival curves according to the intent of surgery and pathologic response at the primary lesion. (**A**) Overall survival according to the intent of surgery after using the targeted agent immediately before surgery. (**B**) Treatment failure-free survival according to the intent of surgery after using the targeted agent immediately before surgery. (**C**) Overall survival according to the pathologic response at the primary lesion after using the targeted agent immediately before surgery. (**D**) Treatment failure-free survival according to the pathologic response at the primary lesion after using the targeted agent immediately before surgery.

**Figure 6 cancers-13-02549-f006:**
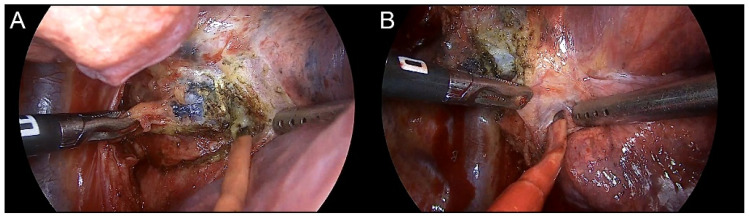
Dense fibrosis or calcification around the initial population cancer cells because of the response to TKI therapy. (**A**) Subcarinal lymph node with proven metastasis confirmed by endobronchial ultrasound biopsy. Residual metastatic lesions were identified in the pathologic report. (**B**) Interlobar lymph node with suspicious metastases revealed using preoperative PET-CT. Complete remission was reported in the final pathologic report.

**Table 1 cancers-13-02549-t001:** Patient characteristics.

Characteristic	*N* or Median	% or IQR
Age at operation, years	59	53.8–65.3
Sex		
Male	15	34.1%
Female	29	65.9%
Smoking history		
Never smoked	31	70.5%
Former smoker	12	27.3%
Current smoker	1	2.3%
Smoking, pack-years	0	0–19.3
ECOG		
0	36	81.8%
1	8	18.2%
PFT		
FEV1, L	2.27	2.0–2.7
FEV1, % predicted	92.0%	83.5–101.0%
DLCO, mL/mmHg/min	17.1	15.4–19.6
DLCO, % predicted	93.0%	77.0–100.0%
Primary tumor location		
RUL	10	22.7%
RML	3	6.8%
RLL	14	31.8%
LUL	6	13.6%
LLL	11	25.0%
Clinical stage at diagnosis ^a^		
IIIB	4	9.1%
IIIC	1	2.3%
IVA	15	34.1%
IVB	24	54.5%
Metastasis sites at diagnosis		
Brain	10	22.7%
Bone	6	13.6%
Lymph nodes	6	13.6%
Intrathoracic organs	5	11.4%
Multiple organs	17	38.6%
Number of distant metastasis at diagnosis		
0	5	11.4%
1–2	12	27.3%
3–5	10	22.7%
>5	17	38.6%
Combined brain metastasis at diagnosis	22	50.00%
Mutational profile at diagnosis		
*EGFR* mutation	32	72.7%
*ALK* fusion	11	25.0%
*ROS1* fusion	1	2.3%

^a^ as per the 8th edition of the TNM classification of lung cancer.

**Table 2 cancers-13-02549-t002:** Preoperative disease status and treatment.

Variables	*N* or Median	% or IQR
Interval from the diagnosis to operation, months	10.9	8.5–18.2
Interval from the initiation of TKI immediately before surgery to the actual operation, months	9.8	7.8–12.9
TKI before surgery		
Gefitinib	15	34.1%
Afatinib	4	9.1%
Erlotinib	1	2.3%
Osimertinib	12	27.3%
- First-line	6	13.6%
- Later-line based on the acquired T790M mutation	6	13.6%
Alectinib	7	15.9%
- First-line	5	11.4%
- Later-line	2	4.5%
Crizotinib	4	9.1%
Lorlatinib	1	2.3%
Best response to TKI immediately before surgery		
Partial response	41	93.2%
Stable disease	3	6.8%
Preoperative local control for metastatic site	21	47.7%
Surgery	4	9.1%
- Brain	3	6.8%
- Lymph node	1	2.3%
Radiation therapy	15	34.1%
- Brain	8	18.2%
- Bone	5	11.4%
- Brain and bone	2	4.5%
Surgery with radiation therapy	2	4.5%
- Brain	2	4.5%
Preoperative number of distant metastases		
0	22	50.0%
1–2	9	20.5%
3–5	6	13.6%
>5	7	15.9%
Preoperative primary lesion status		
Partial response	6	13.6%
Stable disease	30	68.2%
Progressive disease	8	18.2%
Preoperative metastatic lesion status		
Radiologic “No-evidence-of-disease” status	19	43.2%
Partial response	2	4.5%
Stable disease	21	47.7%
Progressive disease	2	4.5%
Intent of surgery		
Residual lesions	34	77.3%
Progressive lesions	10	22.7%

**Table 3 cancers-13-02549-t003:** Operative Outcomes.

Variables	*N* or Median	% or IQR
Extent of pulmonary resection		
Sub-lobar resection	5	11.4%
Lobectomy	37	84.1%
Bi-lobectomy	2	4.5%
Surgical approach		
Thoracotomy	2	4.5%
VATS	37	84.1%
Conversion to thoracotomy	5	11.4%
- Difficult to dissect lymph node	3	6.8%
- Difficult to dissect pulmonary artery	2	4.5%
Combined surgical procedure	20	45.5%
Neck lymph node dissection	7	15.9%
Abdominal lymph node dissection	1	2.3%
Separate pulmonary wedge resection	4	9.1%
En-bloc resection of adjacent lobe	5	11.4%
Other procedure	3	6.8%
Duration of surgery, min	108.5	92.0–136.3
Duration of anesthesia, min	160.0	135.0–191.3
Estimated blood loss, mL		
Minimal (≤50)	29	65.9%
50–200	12	27.3%
>200	3	6.8%
Complete resection (surgical field)		
R0	41	93.2%
R1	1	2.3%
R2	2	4.5%
Intensive care unit stay (duration)	1 (1 day)	2.3%
Chest tube duration, days	4	3.0–5.3
Postoperative hospital stays, days	5	4.0–7.3
Complication (all)	15	34.1%
Prolonged air leak (>5 days)	5	11.4%
Chyle leakage	3	6.8%
Vocal cord palsies	2	4.5%
Bronchopleural fistula	1	2.3%
Acute kidney injury	1	2.3%
Acute lung injury	1	2.3%
Pneumonia	1	2.3%
Pneumothorax	1	2.3%
Complication (Clavien–Dindo > Gr 3)	4	9.1%
Chyle leakage	1	2.3%
Vocal cord palsies	1	2.3%
Broncho-pleural fistula	1	2.3%
Pneumothorax	1	2.3%
In-hospital mortality	0	0.0%

**Table 4 cancers-13-02549-t004:** Pathology and postoperative disease status.

Variables	*N* or Median	% or IQR
Postoperative ypStage		
CR	4	9.1%
I	11	25.0%
II	2	4.5%
III	6	13.6%
IV	21	47.7%
Primary site pathologic response		
CR	5	11.4%
Major pathologic response (≤10%)	10	22.7%
Incomplete pathologic response (>10%)	29	65.9%
Postoperative ypT-stage		
CR	5	11.4%
1	23	52.3%
2	7	15.9%
3	5	11.4%
4	4	9.1%
Postoperative ypN-stage		
0	22	50.0%
1	6	13.6%
2	16	36.4%
Additional mutations in surgical specimens	12	27.3%
T790M	11	25.0%
MET+	1	2.3%
Targeted agent change based on additional mutations in surgical specimens	9	20.5%
Postoperative disease status		
Radiologic “no-evidence-of-disease” status	23	52.3%
Radiologic residual disease	21	47.7%

## Data Availability

The study did not report any data.

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
