# Peer review of "Genetic Analysis and Operative Outcomes in Patients with Oncogene-Driven Advanced NSCLC Treated with Cytoreductive Surgery as a Component of Local Consolidative Therapy"

_cancers, 2021, doi:10.3390/cancers13112549_

Round 1

Reviewer 1 Report

The manuscript, “Genetic Analysis and Operative Outcomes in Oncogene-Driven Advanced NSCLC Patients Treated with Cytoreductive Surgery as a Component of Local Consolidative Therapy.” By Park et al. is a retrospective study of 44 patients, who underwent cytoreductive surgery; demonstrating that pulmonary resection for advanced NSCLC of patients under constant TKI therapy is feasible. Despite of substantial operative morbidities of treatment [15 cases (34.1%)], there was also a substantially improved efficacy of treatment. 5 patients (11.4%) had complete responses and 10 patients (22.7%) showed “major pathologic responses” at the primary lesion.

Unfortunately, in its current form, the study remains anecdotal. There is no comparison to patient control cohorts, even be it derived from published clinical data. Also, there is no comparison in efficacy with alternative treatments, such irradiation combined with TKI therapy.

Thus, taken together, while this study nicely demonstrates that cytoreductive surgery combined with TKI therapy is feasible, remains the question to which extend such a study enhanced our understanding of how best to treat advanced NSCLC.

Author Response

Thanks for your kind comments. We agree and are grateful to you for sharing this important citation. 

We added comments on the points you indicated in the first paragraph of the Discussion as follows.

“Our findings show outstanding results compared to the recently reported study that conducted EGFR-TKI plus thoracic stereotactic body radiation therapy in patients with advanced EGFR-mutant NSCLC. [17]”

[17] Wang, X.; Zeng, Z.; Cai, J.; Xu, P.; Liang, P.; Luo, Y.; Liu, A. Efficacy and acquired resistance for EGFR-TKI plus thoracic SBRT in patients with advanced EGFR-mutant non-small-cell lung cancer: a propensity-matched retrospective study. BMC Cancer 2021, 21, 482, doi:10.1186/s12885-021-08228-2.

Reviewer 2 Report

In this manuscript, the authors describe their findings regarding the outcome regarding response in the surgical specimen and clinical follow-up of neoadjuvant treatment with TKIs in driver mutation positive NSCLC. It describes a relatively small and genomically, therapeutically and surgically varied cohort, which has its limitations regarding statistical analysis and extrapolation to a potential future ITT population. Nonetheless, it provides important information partly due to the fact that the surgical specimens offer information on the pathological response, which is not afforded by other studies describing patients treated with radiotherapy. There are some issues that may be worth addressing though:

  1. It may be difficult to compare the spectrum of mutations found by pcr in the pre-operative specimen to the post-operative NGS. To which extent do the targets in EGFR and the detectable ROS1 fusions overlap, and what is therefore the predicted sensitivity of the pcr based tests bases on primer design and coverage respectively?  
  2. How were the ALK and ROS1 fusions, detected by IHC and PCR respectively, confirmed in the resection specimens? Is this possible with the TSO500 panel?  
  3. On page 4, the authors mention that 93.2% of patients had PR and 6.8% SD. How does this sit with the data in table 2, in which PD is also present?
  4. How was MPR and CR defined in resection specimens? The assessment protocol should be provided or referred to.
  5. On page 8, it is stated that in n=6 cases second-line osimertinib treatment was initiated prior to surgery - is this based on biopsy mutation analysis results (presumably, as this number is also mentioned din fig.3)? If so, it should be stated here.
  6. On page 14, in the limitations paragraph, a liquid biopsy test is mentioned, but details have not been provided as to the outcome.

Author Response

Thank you for kind comments. We addressed all of your comments.

1. It may be difficult to compare the spectrum of mutations found by pcr in the pre-operative specimen to the post-operative NGS. To which extent do the targets in EGFR and the detectable ROS1 fusions overlap, and what is therefore the predicted sensitivity of the pcr based tests bases on primer design and coverage respectively?

The single-gene tests (EGFR, ALK, ROS1) were performed on pre-operative biopsy samples, and NGS for 523 genes, including EGFR, ALK, and ROS1, was performed on post-operative surgical specimens. The single-gene test of EGFR and ROS1 can sensitively detect the most common types. All genetic variations found on a single-gene test before surgery were found on the NGS test after surgery. The NGS test further detected gene mutations associated with targeted drug resistance and co-occurring mutations. Therefore, in this study, the analytical sensitivity of the PCR-based test can be said to be 100% based on the oncogenic (driver) mutation in the preoperative sample. However, this study has limitations as it was not intended to compare single genetic testing (PCR) to NGS directly.

2. How were the ALK and ROS1 fusions, detected by IHC and PCR respectively, confirmed in the resection specimens? Is this possible with the TSO500 panel?

ALK was detected by VENTANA ALK D5F3 CDx Assay (IHC), and ROS1 fusion was detected using the AmoyDx ROS1 Gene Fusions Detection Kit (RT-PCR). Their detection was also possible by TSO500 panel.

3. On page 4, the authors mention that 93.2% of patients had PR and 6.8% SD. How does this sit with the data in table 2, in which PD is also present?

We apologize for causing confusion regarding responses. ORR of 93.2% means “best” overall response. Table 2 represents status prior to surgery. Thus, we revised the “Baseline Characteristics and Treatments Prior to Surgery section” in the Results and “Table 2” as follows.

  • Forty-one patients (93.2%) had partial response, and three patients (6.8%) had stable dis-ease to TKI immediately prior to surgery according to RECIST criteria.

-> In regard to best responses, 41 patients (93.2%) had partial response, and three patients (6.8%) had stable disease after the commencement of TKI therapy immediately before surgery according to RECIST criteria [14].

  • Initial response of TKI immediately prior to surgery

-> Best response of TKI immediately prior to surgery

4. How was MPR and CR defined in resection specimens? The assessment protocol should be provided or referred to.

Thank you for the comment. In our institute, we have been using the following reference. We added the reference to the “Pathologic Analysis of Surgical Specimens” of the Results as follows.

  • Furthermore, 10 patients (22.7%) achieved major pathologic response (MPR), including 5 patients with complete pathologic response (11.4%), at the primary site regardless of the mutation or treatment regimen.

-> Furthermore, based on the criteria proposed by Travis et al.[15],10 patients (22.7%) achieved a major pathologic response (MPR), including 5 patients with a complete pathologic response (11.4%), at the primary site regardless of the mutation or treatment regimen.

[15] Travis WD, Dacic S, Wistuba I, Sholl L, Adusumilli P, Bubendorf L, et al. IASLC Multidisciplinary Recommendations for Pathologic Assessment of Lung Cancer Resection Specimens After Neoadjuvant Therapy. J Thorac Oncol 2020;15:709-40.

5. On page 8, it is stated that in n=6 cases second-line osimertinib treatment was initiated prior to surgery - is this based on biopsy mutation analysis results (presumably, as this number is also mentioned din fig.3)? If so, it should be stated here.

As you indicated, we revised the “NGS and Mutation Profiles section” of the Results as follows:

  • Out of the six patients treated with second-line osimertinib before surgery, three lost their T790M mutation and one developed EGFR C797G mutation.

-> Out of the six patients treated with second-line osimertinib before surgery based on the acquired T790M mutation in preoperative additional biopsy mutation analysis results, three lost their T790M mutation and one developed the EGFR C797G mutation.

6. On page 14, in the limitations paragraph, a liquid biopsy test is mentioned, but details have not been provided as to the outcome.

We apologize for causing confusion regarding liquid biopsy. As it was mentioned in the “Limitations paragraph of the Discussion”, it means “If we performed liquid biopsy, we might capture tumor heterogeneity, but unfortunately, we did not conduct liquid biopsy”. We revised the sentence as follows:

  • Third, with liquid biopsy test, we captured tumor heterogeneity.

-> Third, we did not perform liquid biopsy, which could help capturing tumor heterogeneity.

Round 2

Reviewer 1 Report

I would like to thank the authors to have added another publication, which describes another therapeutic approach. Nevertheless, similarities and differences of these treatments & additional treatments not yet mentioned in the manuscript, should be discussed in much more depth. Otherwise, it becomes challenging to appreciated the value of the presented data.

Author Response

Thanks for your kind comments. We changed the first paragraph of the Discussion as follows.

“Our group has conducted surgical resection as a local consolidative treatment in advanced NSCLC with driver mutations. In the present study of a limited set of patients with relatively short follow-up periods, the 2-year FFS and OS rates were 70.8% and 95.0%, respectively, whereas median OS and FFS had not been reached yet. The rationale for pulmonary resection and its benefits over consolidative radiotherapy are as follows: 1) TKI-treated patients ultimately develop resistance to the drugs, a condition for which there is no established treatment; 2) primary lung lesion is the most common resistant site; 3) surgery is the most reliable method of tumor removal that enables accurate staging and treatment through mediastinal lymph node dissection; 4) surgical tumor samples provide information about tumor heterogeneity and mutational evolution to guide subsequent treatment.”

-> A recent study in patients with advanced EGFR-mutant NSCLC reported that TKIs plus thoracic stereotactic body radiation therapy (SBRT) significantly extends PFS with tolerable toxicity [17]. They reported median PFS was 19.4 months in the TKIs plus SBRT group. Our group has conducted surgical resection as a local consolidative treatment in advanced NSCLC with driver mutations. In the present study of a limited set of patients with relatively short follow-up periods, the 2-year FFS and OS rates were 70.8% and 95.0%, respectively, whereas median OS and FFS had not been reached yet. Although It is difficult to compare directly out results to the previous reports, our findings show a promising prognosis with tolerable surgical morbidity and tumor samples for genetic analysis. The rationale for pulmonary resection and its benefits over consolidative radiotherapy are as follows: 1) TKI-treated patients ultimately develop resistance to the drugs, a condition for which there is no established treatment; 2) primary lung lesion is the most common resistant site; 3) surgery is the most reliable method of tumor removal that enables accurate staging and treatment through mediastinal lymph node dissection; 4) surgical tumor samples provide information about tumor heterogeneity and mutational evolution to guide subsequent treatment.

[17] Wang, X.; Zeng, Z.; Cai, J.; Xu, P.; Liang, P.; Luo, Y.; Liu, A. Efficacy and acquired resistance for EGFR-TKI plus thoracic SBRT in patients with advanced EGFR-mutant non-small-cell lung cancer: a propensity-matched retrospective study. BMC Cancer 2021, 21, 482, doi:10.1186/s12885-021-08228-2.

Reviewer 2 Report

I think the authors for the detailed answers to my previous queries. It is difficult to retrieve if all, and where these answers led to changes in the text. If this applies to only the segments marked in red, this would be insufficient.  

Author Response

Thanks for your kind comments. We addressed all of your comments.

Q1. It may be difficult to compare the spectrum of mutations found by pcr in the pre-operative specimen to the post-operative NGS. To which extent do the targets in EGFR and the detectable ROS1 fusions overlap, and what is therefore the predicted sensitivity of the pcr based tests bases on primer design and coverage respectively?

  • The single-gene tests (EGFR and ROS1) were performed on pre-operative biopsy samples. And for EGFR-mutant patients, NGS for 523 genes, including EGFR and ROS1, and single-gene tests for EGFR were performed on post-operative surgical specimens. The single-gene test to detect EGFR mutations was peptide nucleic acid-mediated real-time PCR-based assay using the PANA Mutyper EGFR Kit (PANAGENE, Daejeon, Korea). ROS1 fusion was detected using the AmoyDx ROS1 Gene Fusions Detection Kit (RT-PCR).

The PANAMutyper test combines PNA-based PCR clamping (PNAClamp) with multiplex fluorescence melting curve analysis (PANAS-Melting) using a fluorescence-labeled PNA probe, which allows detection of 47 hotspot mutations between EGFR exon 18 and exon 21. This method was approved by the Korea Ministry of Food and Drug Safety and highly concordant with the U.S FDA-approved assay (Roche Cobas v2 EGFR assay) validated via several publications [29, 31].

Even though the primers used in the PANAMutyper test do not include C797X in EGFR exon 20 and some of EGFR exon 20 insertions, they include sensitizing mutations (19del and L858R), resistant mutation (T790M), and major uncommon EGFR mutations (S768I, G719X, and L861Q). Based on the incidences of each type of EGFR mutation, it can cover almost 95 to 99% of EGFR mutations with clinical significance. And thus, in this study, the analytical sensitivity of the PCR-based test can be said to be 100% based on the oncogenic (driver) mutation in the preoperative sample.  

The AmoyDx ROS1 Gene Fusions Detection Kit can detect the fusion partner of CD74, EZR, SLC34A2, SDC4, GOPC, TPM3, and LRIG3, which approximately accounts for 83% of ROS1 gene fusions [30].

  • Based on your comments, we added these in the “DISCUSSION” section.
    "Regarding the diagnostic accuracy, the methods that we used do not capture every oncogenic alteration in EGFR mutation and ROS1 fusion. However, detecting EGFR mutations, the PANAMutyper assay can capture sensitizing mutations (19del and L858R), resistant mutation (T790M), and major uncommon EGFR mutations (S768I, G719X, and L861Q) and highly concordant to the Roche cobas® EGFR version 2, which is approved by the U.S. Food and Drug Administration [29]. In addition, ROS1 fusion genes detectable by the AmoyDx ROS1 Gene Fusions Detection Kit account for approximately 83% of all ROS1 gene fusions [30]. However, considering that all of the patients in the study had oncogenic genetic alteration at baseline, the analytical sensitivity of the PCR-based test can be said to be 100% based on the oncogenic (driver) mutation in the preoperative sample."
    [29] Kim, J.O.; Shin, J.Y.; Kim, S.R.; Shin, K.S.; Kim, J.; Kim, M.Y.; Lee, M.R.; Kim, Y.; Kim, M.; Hong, S.H.; et al. Evaluation of Two EGFR Mutation Tests on Tumor and Plasma from Patients with Non-Small Cell Lung Cancer. Cancers (Basel) 2020, 12, doi:10.3390/cancers12040785.

[30] Wu, Y.L.; Yang, J.C.; Kim, D.W.; Lu, S.; Zhou, J.; Seto, T.; Yang, J.J.; Yamamoto, N.; Ahn, M.J.; Takahashi, T.; et al. Phase II Study of Crizotinib in East Asian Patients With ROS1-Positive Advanced Non-Small-Cell Lung Cancer. J Clin Oncol 2018, 36, 1405-1411, doi:10.1200/jco.2017.75.5587.

[31] Han, J.Y.; Choi, J.J.; Kim, J.Y.; Han, Y.L.; Lee, G.K. PNA clamping-assisted fluorescence melting curve analysis for detecting EGFR and KRAS mutations in the circulating tumor DNA of patients with advanced non-small cell lung cancer. BMC Cancer 2016, 16, 627, doi:10.1186/s12885-016-2678-2.

Q2. How were the ALK and ROS1 fusions, detected by IHC and PCR respectively, confirmed in the resection specimens? Is this possible with the TSO500 panel?

  • ALK was detected by VENTANA ALK D5F3 CDx Assay (IHC), and ROS1 fusion was detected using the AmoyDx ROS1 Gene Fusions Detection Kit (RT-PCR). Their detection was also possible by TSO500 panel.
  • As you mentioned, we added this information to the manuscript in the METHOD section, “2.2 Molecular Analyses” part

    “To detect EGFR mutations, peptide nucleic acid-mediated real-time PCR was per-formed using the PANA Mutyper EGFR Kit (PANAGENE, Daejeon, Korea). To identify ALK rearrangements, the VENTANA ALK (D5F3) CDx Assay (Ventana Medical Systems, Tucson, AZ, USA) was performed. ROS1 rearrangements were detected with real-time PCR using the ROS1 Gene Fusions Detection Kit (AmoyDx, Xiamen, China).”
    -> To detect EGFR mutations, peptide nucleic acid-mediated real-time PCR was per-formed using the PANA Mutyper EGFR Kit (PANAGENE, Daejeon, Korea). To identify ALK rearrangements, the VENTANA ALK (D5F3) CDx Assay (Ventana Medical Systems, Tucson, AZ, USA) was performed. ROS1 rearrangements were detected with real-time PCR using the ROS1 Gene Fusions Detection Kit (AmoyDx, Xiamen, China). Their detection was also possible by next-generation sequencing, which will be described as follows.

Q3. On page 4, the authors mention that 93.2% of patients had PR and 6.8% SD. How does this sit with the data in table 2, in which PD is also present?

  • We apologize for causing confusion regarding responses. ORR of 93.2% means “best” overall response. Table 2 represents status prior to surgery. Thus, we revised the “Baseline Characteristics and Treatments Prior to Surgery section” in the Results and “Table 2” as follows.
  • “Forty-one patients (93.2%) had partial response, and three patients (6.8%) had stable disease to TKI immediately prior to surgery according to RECIST criteria.”

-> In regard to best responses, 41 patients (93.2%) had partial response, and three patients (6.8%) had stable disease after the commencement of TKI therapy immediately before surgery according to RECIST criteria [15]. Among them, eight and two patients showed disease progression in the primary site and metastatic sites, respectively.

  • In Table 2
    “Initial response of TKI immediately prior to surgery”

-> Best response of TKI immediately prior to surgery

Q4. How was MPR and CR defined in resection specimens? The assessment protocol should be provided or referred to.

  • Thank you for the comment. In our institute, we have been using the following reference [15]. We added the definition of MPR and CR and referenced it In the Method section as follows.
  • "Based on the criteria proposed by Travis et al. [15], MPR (major pathologic response) is defined historically as 10% or less residual viable tumor following preoperative treatment, and CR (complete pathologic response) is defined as no viable tumor following preoperative treatment."

[15] Travis WD, Dacic S, Wistuba I, Sholl L, Adusumilli P, Bubendorf L, et al. IASLC Multidisciplinary Recommendations for Pathologic Assessment of Lung Cancer Resection Specimens After Neoadjuvant Therapy. J Thorac Oncol 2020;15:709-40.

Q5. On page 8, it is stated that in n=6 cases second-line osimertinib treatment was initiated prior to surgery - is this based on biopsy mutation analysis results (presumably, as this number is also mentioned din fig.3)? If so, it should be stated here.

  • As you indicated, we revised the “NGS and Mutation Profiles section” of the Results as follows:
  • "Out of the six patients treated with second-line osimertinib before surgery, three lost their T790M mutation and one developed EGFR C797G mutation."

-> Out of the six patients treated with second-line osimertinib before surgery based on the acquired T790M mutation in preoperative additional biopsy mutation analysis results, three lost their T790M mutation and one developed the EGFR C797G mutation.

Q6. On page 14, in the limitations paragraph, a liquid biopsy test is mentioned, but details have not been provided as to the outcome.

  • We apologize for causing confusion regarding liquid biopsy. As it was mentioned in the “Limitations paragraph of the Discussion”, it means “If we performed a liquid biopsy, we might capture tumor heterogeneity, but unfortunately, we did not conduct liquid biopsy”. We revised the sentence as follows:
  • "Third, with liquid biopsy test, we captured tumor heterogeneity."

-> Third, we did not perform a liquid biopsy before and after the TKIs. If performed, it could help to capture tumor heterogeneity.